# Eltrombopag Improves Erythroid Differentiation in a Human Induced Pluripotent Stem Cell Model of Diamond Blackfan Anemia

**DOI:** 10.3390/cells10040734

**Published:** 2021-03-26

**Authors:** Husam Qanash, Yongqin Li, Richard H. Smith, Kaari Linask, Sara Young-Baird, Waleed Hakami, Keyvan Keyvanfar, John S. Choy, Jizhong Zou, Andre Larochelle

**Affiliations:** 1Cellular and Molecular Therapeutics Branch, National Heart, Lung and Blood Institute (NHLBI), National Institutes of Health (NIH), Bethesda, MD 20892, USA; 26qanash@cua.edu (H.Q.); yongqin.li@nih.gov (Y.L.); smithr@nhlbi.nih.gov (R.H.S.); waleed.hakami@nih.gov (W.H.); 2Department of Biology, Catholic University of America, Washington, DC 20064, USA; choy@cua.edu; 3Department of Medical Laboratory Science, College of Applied Medical Sciences, The University of Hail, Hail 55476, Saudi Arabia; 4iPSC Core Facility, NHLBI, NIH, Bethesda, MD 20892, USA; linaskk@nhlbi.nih.gov (K.L.); jizhong.zou@nih.gov (J.Z.); 5Eunice Kennedy Shriver, National Institute of Child Health and Human Development, NIH, Bethesda, MD 20892, USA; sara.young@nih.gov; 6National Institute of General Medical Sciences (NIGMS), NIH, Bethesda, MD 20892, USA; 7Department of Medical Laboratories Technology, College of Applied Medical Sciences, Jazan University, Jazan 45142, Saudi Arabia; 8Clinical Flow Core Facility, NHLBI, NIH, Bethesda, MD 20892, USA; keyvanfk@nhlbi.nih.gov

**Keywords:** diamond blackfan anemia, induced pluripotent stem cells, eltrombopag, genome editing, disease modeling, drug testing

## Abstract

Diamond Blackfan Anemia (DBA) is a congenital macrocytic anemia associated with ribosomal protein haploinsufficiency. Ribosomal dysfunction delays globin synthesis, resulting in excess toxic free heme in erythroid progenitors, early differentiation arrest, and pure red cell aplasia. In this study, DBA induced pluripotent stem cell (iPSC) lines were generated from blood mononuclear cells of DBA patients with inactivating mutations in RPS19 and subjected to hematopoietic differentiation to model disease phenotypes. In vitro differentiated hematopoietic cells were used to investigate whether eltrombopag, an FDA-approved mimetic of thrombopoietin with robust intracellular iron chelating properties, could rescue erythropoiesis in DBA by restricting the labile iron pool (LIP) derived from excessive free heme. DBA iPSCs exhibited RPS19 haploinsufficiency, reduction in the 40S/60S ribosomal subunit ratio and early erythroid differentiation arrest in the absence of eltrombopag, compared to control isogenic iPSCs established by CRISPR/Cas9-mediated correction of the RPS19 point mutation. Notably, differentiation of DBA iPSCs in the presence of eltrombopag markedly improved erythroid maturation. Consistent with a molecular mechanism based on intracellular iron chelation, we observed that deferasirox, a clinically licensed iron chelator able to permeate into cells, also enhanced erythropoiesis in our DBA iPSC model. In contrast, erythroid maturation did not improve substantially in DBA iPSC differentiation cultures supplemented with deferoxamine, a clinically available iron chelator that poorly accesses LIP within cellular compartments. These findings identify eltrombopag as a promising new therapeutic to improve anemia in DBA.

## 1. Introduction

Diamond Blackfan Anemia (DBA) is a dominantly inherited bone marrow failure syndrome primarily characterized by defective erythropoiesis. In more than half of patients with DBA, heterozygous mutations have been identified in genes encoding ribosomal proteins (RP), resulting in RP haploinsufficiency and abnormal ribosome biogenesis or function [1,2]. RPS19 is the most commonly mutated gene in DBA, and monoallelic silencing of RPS19 results in aberrant assembly of the small 40S ribosomal subunits. In select subjects with DBA, hematopoietic stem and progenitor cell (HSPC) transplantation can offer a potential cure. For most patients, corticosteroids and long-term transfusion support remain the mainstay of treatment. However, steroid toxicity precludes long-term use and acquired transfusional hemochromatosis may have serious clinical sequelae. Novel drug candidates with therapeutic potential in DBA, including the translation enhancer L-leucine [3,4,5], the calmodulin inhibitor trifluoperazine [6] and the transforming growth factor-beta (TGF-β) receptor inhibitor sotatercept [7,8,9], have entered clinical trials but there have been no reports of efficacy in cohorts of DBA patients to date. Thus, there is a considerable need for alternative therapeutic modalities in this disease.

Erythropoiesis halts at or just before the proerythroblast stage in DBA patients. A stoichiometric imbalance between heme and globin has been proposed to explain the early termination of red blood cell (RBC) differentiation in DBA [10,11,12]. In this model, heme synthesis proceeds normally in erythroid progenitors (burst/colony forming unit erythroid, BFU-E/CFU-E) and proerythroblasts, whereas globin translation is delayed due to ribosomal dysfunction. Free heme is thus in excess of globin in these cells. Excessive free heme induces cell death through apoptosis and ferroptosis, an iron-dependent process associated with increased reactive oxygen species (ROS) and unchecked lipid peroxidation. Recently, excess free heme in the early phase of RBC differentiation has also been shown to induce premature downregulation of GATA1, the primary regulator of erythroid differentiation, contributing to early termination of erythropoiesis [10]. These observations suggest that limiting heme synthesis might improve erythropoiesis in DBA. Iron restriction provides a clinically relevant approach to limit heme production. The iron chelators deferasirox (DFX) and deferoxamine (DFO) have been used extensively in the clinic to reduce iron overload in patients requiring red cell transfusions long-term. While administration of DFX and DFO is generally well tolerated, including in subjects with DBA [13], costs associated with prolonged use may be prohibitive and potentially serious, sometimes fatal, consequences have been documented, including gastrointestinal hemorrhage, cytopenias and hepato-renal toxicity [14].

Eltrombopag (EPAG) is a synthetic orally bioavailable small molecule mimetic of thrombopoietin (TPO), a key regulator of platelet production and HSPC maintenance. It is approved for the treatment of cytopenias in patients with severe aplastic anemia (SAA) [15,16], and of thrombocytopenia associated with chronic immune thrombocytopenic purpura [17] and chronic hepatitis C [18]. The hematopoietic effects of EPAG primarily depend on its interaction with the transmembrane domain of the TPO receptor (c-mpl) and activation of downstream signaling cascades in target cells [19,20,21]. EPAG also displays antiproliferative properties independent of the TPO receptor in cancer cell culture models [22,23,24,25]. These anticancer effects are elicited in part by the molecule’s high-affinity iron binding properties. EPAG was shown to efficiently reduce the labile iron pool (LIP), a redox-active chelatable iron pool within cells, leading to decrements in iron-induced ROS and restoration of cellular function in ex vivo culture systems [26]. Notably, when compared with DFX, EPAG could remove intracellular iron with higher efficiency in human cell lines [26].

In this study, we hypothesized that the iron-chelating properties of EPAG could restrict iron availability and limit heme synthesis in DBA erythroid progenitors, thereby reestablishing the heme-globin ratio and promoting erythroid maturation. To test this possibility, we chose a reprogramming approach to generate induced pluripotent stem cells (iPSCs) from DBA patients with RPS19 mutations. DBA iPSCs can be readily expanded in culture to provide a renewable source of cells for investigation, and erythroid cells differentiated from DBA iPSCs have previously been shown to recapitulate in vitro the pathological features of the disease [27,28,29]. Protocols primarily based on embryoid body (EB) formation have been utilized in previous studies to facilitate the emergence of human hematopoietic cells from DBA iPSCs but the scale-up of these approaches relies on complex transcription factor-based reprogramming systems to overcome the intrinsic inefficiencies of EB-based iPSC differentiation [29,30]. Here, we applied an alternative differentiation approach to generate erythroid cells from human DBA iPSCs based on a simple, scalable, monolayer platform we previously developed for hematopoietic differentiation of human iPSCs derived from healthy individuals [31]. Importantly, we find that addition of EPAG during DBA iPSC differentiation significantly improves the erythroid maturation defect recapitulated in this model, suggesting that EPAG may show clinical efficacy in patients with DBA.

## 2. Materials and Methods

### 2.1. Generation and Culture of DBA iPSCs

Mononuclear cells (MNCs) from three DBA donors (Table 1) were obtained after informed consent, in accordance with the Declaration of Helsinki, under an Institutional Review Board-approved clinical protocol (NCT00027274). DBA MNCs from each donor were reprogrammed into iPSCs using the integration-free CytoTune Sendai viral vector kit (A16517, Thermo Fisher Scientific, Waltham, MA, USA) following a method previously reported [32,33]. Induced PSCs were maintained in 6-well tissue culture plates coated with Matrigel (354230, Corning Life Sciences, Tewksbury, MA, USA) in Essential 8™ (E8) Medium (A1517001, Thermo Fisher Scientific). Culture medium was changed daily and iPSCs were split every three to four days. Briefly, E8 medium was aspirated, and wells were sequentially washed with phosphate buffered saline (PBS) or 0.5 mM EDTA in PBS (PBS/EDTA). Cells were then dissociated in PBS/EDTA 0.5 mM for 2–3 min and pelleted by centrifugation. PBS/EDTA was aspirated and replaced with 1 mL E8 medium containing 1.25 µM ROCK inhibitor (Y0503, Sigma-Aldrich Corp, St. Louis, MO, USA). Cells were pipetted 3–4 times using a P1000 pipette to dissociate into small to medium sized clusters and split onto new 6-well plates at various dilutions. Cells were characterized by alkaline phospha tase staining, immunophenotyping, karyotyping, teratoma assay, immunohistochemistry, Western blotting and polysome profiling.

### 2.2. Alkaline Phosphatase Staining

Alkaline phosphatase staining was performed per manufacturer’s instructions (SK-5400, Vector Laboratories, Burlingame, CA, USA). The reprogramming efficiency was calculated by dividing the number of iPSC colonies counted in three wells of a 6-well plate by the number of cells plated on the day of transduction with Sendai vectors.

### 2.3. Immunophenotyping of iPSCs

iPSCs were harvested as detailed above and stained with Tra-1-60 (MA1-023, Thermo Fisher Scientific) and Nanog (FCABS352A4, Millipore Sigma, St. Louis, MO, USA) antibodies following manufacturer’s instructions (Appendix A). Flow cytometry was performed on an LSRII Fortessa analyzer (Becton Dickinson, Franklin Lakes, NJ, USA). All flow data were analyzed using FlowJo 10.6 Software (FlowJo LLC).

### 2.4. Karyotyping Analysis

DBA MNCs-derived iPSCs were maintained and grown until they reached 70–80% confluency. Cells were then collected and karyotyped based on the GTW-banded chromosome method at the WiCell Research Institute (Madison, WI, USA). Twenty metaphase cells were analyzed for each iPSC clone.

### 2.5. Teratoma Assay

Approximately 1 × 10^7^ iPSCs were harvested using 0.5 mM EDTA in PBS and then resuspended in 500 µL of E8 medium supplemented with 25 mM HEPES (pH 7.4). After cooling on ice, 250 µL (50% Vol) of cold Matrigel was mixed in and then quickly injected subcutaneously into the groin areas of NOD-scid-IL2Rgnull (NSG) mice (Stock # 005557, Jackson Laboratory, Bar Harbor, ME, USA) at 150 µL per injection site. Typically, each cell line was injected into two NSG mice at four separate injection sites. Eight to twelve weeks after injection, tumors were harvested from euthanized mice and then fixed in 10% Neutral Buffered Formalin. Paraffin sections were prepared from each tumor and stained with hematoxylin and eosin (H&E). Animals were housed and handled in accordance with the guidelines set by the Committee on Care and Use of Laboratory Animals of the Institute of Laboratory Animal Resources, National Research Council (DHHS publication No. NIH 85–23), and the protocol was approved by the Animal Care and Use Committee of the NHLBI.

### 2.6. Immunohistochemistry

Immunohistochemistry for detection of Sendai vector (SeV)-derived antigens in DBA iPSCs was conducted as previously described [34,35,36,37]. Briefly, DBA iPSCs were passaged and seeded onto Matrigel chamber slides for 24 h to allow complete attachment. They were washed once with 5% BSA/PBS for 10 min and fixed with 4% formaldehyde (1004960700, Millipore Sigma) for 5 min at room temperature (RT). Cells were then rinsed twice and incubated with 1:500 primary rabbit anti-SeV antibody (PD029, MBL International Corporation, Woburn, MA, USA) for 1 h at 37 °C. Then, cells were rinsed three times and incubated with 1:1000 goat anti-rabbit IgG horseradish peroxidase (HRP) conjugated-second-antibody (A11034, Invitrogen, Grand Island, NY, USA) for 1 h at 37 °C. The secondary antibody was removed, and cells were washed twice with 5% BSA/PBS and once with nuclear stain DAPI/5% BSA/PBS. Cells were visualized under a total internal reflection fluorescence microscope.

### 2.7. Western Blot

Western blot analysis was performed as previously described [38]. Briefly, DBA, iso-genic, and WT iPSCs derived from a healthy donor were washed with PBS (25-507B, Genesee Scientific, San Diego, CA, USA) three times and collected with radioimmunoprecipitation assay-cell lysis buffer (89900, Thermo Fisher Scientific). Cell lysates were separated via 10% Bis-Tris polyacrylamide gel electrophoresis (PAGE) (NP0315, Thermo Fisher Scientific) with MES SDS running buffer (NP0002, Thermo Fisher Scientific) combined with NuPAGE antioxidant (NP0005, Thermo Fisher Scientific). Gel bands were then electro-transferred to PVDF nitrocellulose membranes (PB5210, Thermo Fisher Scientific). Membranes were blocked with Blotto, non-fat dry milk (sc-2324, Santa Cruz Biotechnology, Santa Cruz, CA, USA) for 1 h at 4 °C, followed by incubation with 1:1000 primary rabbit anti-RPS19 antibody (Ab155994, Abcam, Cambridge, MA, USA) overnight at 4 °C. Blots were washed three times with Tris-buffered saline with Tween-20 (TBST, 28352, Thermo Fisher Scientific) and incubated with 1:2000 goat anti-rabbit HRP linked secondary antibody (7074, Cell Signaling Technology, Danvers, MA, USA) for 60–90 min at 4 °C. For the housekeeping control protein (Tubulin), membranes were restored by incubation in stripping buffer (2059, Thermo Fisher Scientific) for 30 min at 42 °C. Blots were then incubated in blocking buffer (5% milk) for 1 h at RT, followed by incubation with 1:2000 primary mouse anti-Tubulin antibody (MAB3408, Millipore Sigma) overnight at 4 °C. Membranes were rinsed and incubated with 1:5000 goat anti-mouse IgG HRP-conjugated secondary antibody (31432, Thermo Fisher Scientific) for 1.5 h at 4 °C. Following three washes with TBST to remove excess antibody, membranes were soaked for 3 min in SuperSignal™ West Dura Extended Duration Substrate (34076, Thermo Fisher Scientific) and then analyzed via chemiluminescence using an Amersham Imager 680 (Model 29270772, GE Healthcare Life Sciences, Marlborough, MA, USA).

### 2.8. Polysome Profiling

Polysome profiling was conducted as previously described [39]. DBA and isogenic iPSCs and iPSCs derived from a healthy donor were grown in a 150 mm tissue culture plates and treated with 50 µg/mL of cycloheximide (CHX) for 10 min immediately prior to lysate collection. Plates were subsequently transferred to ice, and iPSCs were washed with 10 mL 1 × PBS containing 50 µg/mL CHX. Cells were lysed with 0.4 mL lysis buffer (20 mM Tris-HCl, pH 7.5; 100 mM NaCl; 10 mM MgCl_2_; 0.4% NP-40; 50 µg/mL CHX; and 1 complete EDTA-free protease inhibitor cocktail tablet) and sheared by passing the cell lysate through a 1 mL syringe with a 25 gauge needle 10 times. Lysates were clarified by centrifugation for 10 min at 13,000 RPM and 4 °C. The concentration of each lysate was determined, and equal A260 units were gently layered on top of a 10–50% sucrose gradient (20 mM Tris-HCl, pH 7.5; 100 mM NaCl; 10 mM MgCl_2_; 0.4% NP-40; and 50 µg/mL CHX) prepared using a BioComp Gradient Master. Sucrose gradients were ultra-centrifuged, using a SW41 rotor at 40,000 RPM and 4 °C for 2 h, followed by fractionation and collection of whole cell lysate polysome profiles with a BioComp Gradient Fractionator.

### 2.9. Generation of Isogenic iPSCs Using CRISPR/Cas9 Technology

Optimization experiments were initially conducted to identify an optimal DBA sgRNA for the generation of genetically defined isogenic sibling lines from DBA863 iPSCs. Two DBA sgRNAs (sgRNA1 and sgRNA2) were used in this experiment (Appendix A). Approximately 1 × 10^6^ DBA cells were electroporated with 20 µg purified Cas9 protein (PNA Bio, Thousand Oaks, CA, USA) and 10 μg of either sgRNA1 or sgRNA2 using a NEPA21 super electroporator to determine knockout (KO) efficiency. Cells were then maintained in E8 Medium with 10 µM ROCK inhibitor (1254, Tocris Bioscience, Ellisville, MO, USA) for 2–3 days before collection. Genomic DNA extraction per manufacturer’s instructions (69504, Qiagen, Valencia, CA, USA). PCR amplicons were obtained using RPS19-specific forward and reverse primers (Appendix A). PCR parameters were: 95 °C for 3 min, followed by 35–40 cycles of 98 °C for 30 s, 56 °C for 30 s, and 72 °C for 1 min, with a final extension period of 72 °C for 5 min. For T7 endonuclease I (T7EI) assay, the PCR products (~200 ng of amplicon) were mixed with 10X NEBuffer #2 (M0302S, NEB, Ipswich, MA, USA) in 20 µL total volume. PCR amplicons were then heat denatured and re-annealed in a thermocycler using the following conditions: 95 °C for 5 min, ramp from 95 °C to 85 °C at −2 °C/second, ramp from 85 °C to 25 °C at −0.1 °C/second, and then hold at 22 °C. The amplicons then were cleaved using 10 units T7EI (M0302S, NEB) for 15 min at 37 °C to recognize and cleave imperfectly matched DNA, and visualized on 4–20% polyacrylamide TBE gels (EC6225, Invitrogen) using a BioDoc-It^TM^ Imaging System (Model 97-0165-02, UVP BioDoc-It^TM^).

DBA iPSCs were passaged 1:3 into 6-well plates and maintained in E8 medium with 1 × RevitaCell (A2644501, Gibco, Carlsbad, CA, USA) for 1–2 h until transfection. DBA iPSCs were subjected to lipid-mediated transfection by addition of 2.5 mL E8 supplemented with 1 × RevitaCell and 200 µL OptiMEM (31985088, Gibco) containing 5 µL Lipofectamine Stem (STEM00001, Invitrogen), 1 pmol eCas9-EGFP mRNA, 20 pmol DBA sgRNA2 and 20 or 30 pmol R62Q correction ssODN. The latter contained 3 silent mutations to abolish the PAM sequence and create an AscI restriction enzyme digestion site for rapid identification of corrected clones (Appendix A) [40]. GFP^+^ iPSCs were sorted 24–48 h after transfection and re-plated in 96-well plates as cell pools of 20 or 200 cells/well in E8 Flex medium (A2858501, Gibco) supplemented with 1 × CloneR (05888, STEMCELL Technologies, Inc., Vancouver, BC, Canada). Corrected iPSC clones were identified by AscI restriction fragment length polymorphisms (RFLPs) and characterized as described above.

### 2.10. Identification of Corrected iPSC Clones

For identification of corrected iPSC clones by AscI RFLPs, genomic DNA was first isolated using Agencourt DNAdvance Kit (A48705, Beckman Coulter Life Sciences, Indianapolis, IN, USA). PCR was then performed with Phusion^®^ Hot Start Flex DNA Polymerase (M0535L, NEB) for 43 cycles at 60 °C in GC buffer with 4 µM SYTO-82 (S1133, Thermo Fisher Scientific). Two primers (isogenic F and R primers, Appendix A) were used to amplify the area of interest in the RPS19 gene. DNA products were purified with Agencourt AMPure XP (A63881, Beckman Coulter Life Science). Elution volume was adjusted based on SYTO 82 relative fluorescent unit. After an overnight digestion with AscI restriction enzyme (R0558S, NEB), RFLPs were analyzed on a 1% agarose gel with ethidium bromide (54803, Lonza, Walkersville, MD, USA) in Accu GENE^TM^ 1 × Tris-acetate-EDTA buffer (50844, Lonza). AscI positive clones were then subjected to Sanger sequencing to confirm genetic correction.

### 2.11. Hematopoietic Differentiation of DBA and Isogenic iPSCs

iPSCs were differentiated for 21 days using the STEMdiff™ Hematopoietic Kit (05310, STEMCELL Technologies, Inc.), as previously described [31]. Briefly, one day be-fore differentiation (Day −1), iPSCs were split as described above, and cluster concentrations were calculated. A total of 35–40 clusters (>20 cells/cluster) were transferred per well into a Matrigel-coated 12-well plate and cultured overnight. Wells that were either under confluent (<10 adherent clusters) or overconfluent (>50 adherent clusters) were discarded. On Day 0 of differentiation, medium A (containing basic fibroblast growth factor [bFGF], bone morphogenetic protein 4 [BMP4] and vascular endothelial growth factor A [VEGFA]) was added to promote mesodermal differentiation and a half medium A change was done on day 2. On Day 3 of differentiation, supernatant was removed and 1 mL hematopoietic differentiation medium B (containing bFGF, BMP4, VEGFA, stem cell factor [SCF], FMS-line tyrosine kinase 3 ligand [Flt3L] and TPO) was added to each well, followed by half-medium change on day 5, 7, 9, 11, 13, 15, 17, and 19. In some experiments, DBA and isogenic iPSC cultures were supplemented with iron chelators from day 11 to 21 of differentiation, including 3 µg/mL (6.8 µM) EPAG (Novartis, Cambridge, MA, USA), 10 µM DFX (16753, Cayman Chemical, Ann Arbor, MI, USA), or 10 µM DFO (14595, Cayman Chemical), to assess the impact of each drug on erythroid maturation. Concentrations of EPAG, DFX and DFO were selected based on data from preliminary optimization experiments demonstrating the most effective erythroid maturation with limited cellular toxicity.

### 2.12. Collection of Adherent and Non-Adherent Cells Differentiated from iPSCs

Differentiated cells from each well were harvested on day 19 or 21 of iPSC differentiation when maximal erythroid production was observed. Both adherent and non-adherent cells were collected and combined for analysis. Non-adherent cells were harvested first by vigorous pipetting. Adherent cells were washed once with PBS, incubated with Accutase (07920, STEMCELL Technologies, Inc.) for 10 min at 37 °C, and detached by two vigorous washes with PBS/10% FBS (FACS buffer). The resulting cell suspension was filtered using 40 µm cell strainer (431750, Corning Life Sciences), centrifuged at 300× *g* for 5 min and resuspended in 1 mL FACS buffer for counting and downstream assays, including flow cytometry, colony forming unit (CFU) assay and LIP quantification.

### 2.13. Immunophenotyping and Sorting of iPSC-Differentiated Cells

Induced PSC-differentiated cells harvested at day 19 or 21 of differentiation were stained with antibodies (Appendix A) following manufacturer’s instructions, and analyzed on an LSRII Fortessa instrument (Becton Dickinson). All flow data were analyzed using FlowJo 10.6 Software (FlowJo LLC). For CFU assays, iPSC-differentiated cells collected at day 19 or 21 of differentiation and sorted to isolate hematopoietic populations (CD43^+^ and CD45^+/−^), using BD FACSAria™ II or BD FACSAria™ Fusion instruments with a 100 µm nozzle [31].

### 2.14. CFU Assay

Day 19 or 21 sorted hematopoietic cells were collected in Iscove Modified Dulbecco Medium (IMDM)/10% fetal bovine serum (FBS), centrifuged at 300× *g* for 15 min and resuspended in IMDM/2% FBS. Human CFU assays were performed per manufacturer’s instructions using MethoCult medium (04445, STEMCELL Technologies, Inc.). Briefly, 9000 total sorted CD43^+^ and CD45^+/−^ cells were resuspended in 3 mL methylcellulose medium and 3000 cells were plated in duplicate per 35 mm dish for all conditions. Cells were then incubated for 12–16 days at 37 °C prior to the counting and morphological identification of erythroid and myeloid colonies.

### 2.15. Quantification of Intracellular LIPs

Human isogenic DBA iPSCs were subjected to hematopoietic and erythropoietic differentiation for 19 days, harvested and incubated with 250 nM calcein-AM (C1430, Invitrogen) for 10 min at 37 °C. Cells were subsequently washed with FACS buffer and resuspended in prewarmed stem cell differentiation medium B (05310, STEMCELL Technologies, Inc.) with or without 3 µg/mL EPAG, 10 µM DFX, or 10 µM DFO (D9533, Sigma-Aldrich Corp.) for 4 h at 37 °C and 5% CO_2_ [41]. Intracellular calcein-AM fluorescence was measured in erythroid progenitor cells by flow cytometry on an LSRII Fortessa, and mean fluorescence intensity was calculated using FlowJo version 10.6 (FlowJo LLC).

### 2.16. Statistical Analysis

Results were analyzed with GraphPad Prism Software (version 8.4.1), using unpaired Student *t*-tests. All results are presented as mean ± standard error of the mean (SEM) and * signifies *p* < 0.05, ** *p* < 0.01, *** *p* < 0.001, and **** *p* < 0.0001.

## 3. Results

### 3.1. Generation and Characterization of DBA and Isogenic iPSCs

To investigate whether EPAG can improve erythroid maturation in DBA, we first established an iPSC disease model by Sendai vector-mediated overexpression of Oct4, Sox2, Klf4 and c-Myc in MNCs collected from three patients, each with a distinct point mutation within the RPS19 gene (Table 1).

Stable iPSC lines were readily obtained for each subject with reprogramming efficiencies of 0.3–0.6%. All iPSC clones were independently confirmed to have a normal karyotype (Appendix A), lack vector-derived antigens (Appendix A), and display characteristics of pluripotency in flow cytometric (Appendix A) and teratoma assays (Appendix A). Sucrose gradient polysome profiles revealed marked ribosome biogenesis defects, as evidenced by reduced 40S/60S ratios and 80S monosomes in all DBA iPSC lines relative to a wild-type (WT) control iPSC line derived from a healthy donor (Appendix A).

Next, we established genetically defined isogenic sibling iPSC clones to ensure that phenotypes observed in DBA iPSCs are driven by disease-specific mutations and do not result from genetic or epigenetic alterations introduced during reprogramming. To this end, genome editing tools based on CRISPR/Cas9 were optimized and applied to revert the DBA-associated point mutation in one iPSC clone (DBA863, c.185G>A, hereafter referred to as DBA) to a wild-type sequence (Appendix A), as confirmed by Sanger sequencing (Figure 1A). We verified that genetic manipulation did not alter the karyotype or the pluripotent nature of these cells (Appendix A). By Western blotting, we observed haploinsufficiency of total cellular RPS19 protein in DBA iPSCs, and restoration to steady-state levels in isogenic sibling iPSCs, similar to RPS19 protein levels detected in a WT control iPSC clone (Figure 1B). Consistent with genetic and phenotypic correction of RPS19 haploinsufficiency, normalization of ribosome biogenesis was observed in polysome profiles of isogenic iPSCs, as indicated by normalization of the 40S/60S ratio and restoration of the monosome (80S) fraction (Figure 1C,D).

### 3.2. Hematopoietic Differentiation of DBA iPSCs Phenocopies In Vitro the Erythroid Maturation Defect Found in DBA

During normal erythropoiesis, distinct cell surface proteins are sequentially expressed, defining early to late stages of erythroid maturation (Figure 2A). To evaluate whether patient-derived iPSCs could recapitulate in vitro the early block in erythroid differentiation observed in DBA patients, clone DBA863 and its isogenic counterpart were subjected to hematopoietic differentiation using the STEMdiff^TM^ monolayer-based approach [31]. In this system, a supportive CD43^−^CD45^−^ non-hematopoietic adherent monolayer initially forms and, following addition of hematopoietic cytokines, CD43^+^CD45^+/−^ hematopoietic cells emerge from the monolayer before their eventual release in the culture supernatant. We previously described two waves of erythroid development during iPSC differentiation using this approach [31]. The first population peaks early (day 7) and rapidly disappears. The second wave of erythroid cells progressively develops from day 10 through day 21 of differentiation and peaks at day 19–21. Embryonic (ɛ) globins prevail at day 7, whereas red cells derived from later stages of differentiation (day 19 to 21) predominantly express adult-type ɣ and β-globins, reminiscent of successive waves of primitive (first wave) and definitive (second wave) hematopoiesis during embryonic development [31]. Since hematopoietic cells of the primitive wave are transient and remain of unclear relevance to adult hematopoiesis, cells differentiated from DBA and isogenic iPSCs were harvested between day 19 and 21 (wave 2) of culture for further characterization.

Isogenic and DBA iPSCs gave rise to similar percentages of total CD71^+^ erythroid progenitor and precursor cells within the hematopoietic CD43^+^CD45^+/−^ population (Appendix A). To assess the potential of these cells to differentiate beyond the critical block in differentiation at the proerythroblast stage in DBA, we compared the frequency of late stage CD45^−^ erythroblasts co-expressing CD71 and CD235a after differentiation from DBA and isogenic iPSC lines. The control isogenic group efficiently gave rise to mature CD71^+^CD235a^+^ cells with a mean frequency of 11.40 ± 1.9% (Figure 2B,C). In contrast, percentages of CD71^+^CD235a^+^ erythroid cells were significantly lower (2.0 ± 0.3%, *p* = 0.0001) after differentiation of DBA iPSCs (Figure 2B,C). In line with this finding, Giemsa stains of hematopoietic cells differentiated from isogenic iPSCs revealed predominantly mature orthochromatic erythroblasts, whereas morphologically larger early-stage erythroblasts (e.g., proerythroblasts) were more readily identified in the DBA group (Figure 2D). In CFU assays, DBA iPSCs generated numbers of myeloid colonies (CFU-G, CFU-M and CFU-GM) comparable to isogenic iPSCs, but erythroid colonies (BFU-E/CFU-E) were largely undetectable, in keeping with DBA erythroid progenitor’s inability to differentiate in vitro (Figure 2E,F).

### 3.3. Defective Erythropoiesis in DBA iPSCs Is Partially Rescued by EPAG

To address the hypothesis that EPAG may rescue the erythroid defect in DBA by restricting iron availability, we first sought to confirm the intracellular iron chelating properties of EPAG in normal erythroid cells differentiated from isogenic iPSCs using the calcein-AM fluorometric assay. When bound to iron, the fluorescent signal of calcein-AM is quenched; upon addition of an iron chelator, iron is released from calcein-AM and its fluorescence intensifies. As previously reported in various human cell lines [24], iPSC derived erythroid cells loaded with calcein-AM and then treated with EPAG displayed a marked intensification of calcein-AM fluorescent signal compared to untreated controls (Figure 3A,B). Furthermore, we observed reduced erythroid differentiation potential of isogenic iPSCs when EPAG was added to medium for the last 8–10 days of culture (day 11 to 19 or 21) when the second wave of erythropoiesis is observed in this system (Appendix A) [31]. Together, these results support EPAG’s ability to restrict iron availability by mediating intracellular iron chelation in these cells. We next differentiated DBA iPSCs in the presence of EPAG from day 11 of culture. Notably, EPAG improved erythroid maturation, as indicated by increased percentages of more mature CD71^+^CD235a^+^ erythroblastic populations compared to cultures without EPAG (Figure 3C,D). In agreement with this finding, late-stage orthochromatic erythroblasts were readily detected on Giemsa stains of hematopoietic cells differentiated from DBA iPSCs in the presence of EPAG (Figure 3E). Additionally, hematopoietic populations derived from DBA iPSC cultures containing EPAG could form erythroid colonies in CFU assays (Figure 3F,G), albeit at reduced levels compared to isogenic controls (Figure 2E). In contrast, addition of EPAG during DBA iPSC differentiation had no significant impact on non-erythroid CFU output (Figure 3F). Hence, defective erythropoiesis could be rescued at least in part by addition of EPAG during DBA iPSC differentiation, and this effect was independent of EPAG’s activity as regulator of HSPC proliferation and multilineage hematopoiesis.

### 3.4. Intracellular Iron Restriction Improves Erythropoiesis in DBA iPSCs

To assess whether EPAG-mediated intracellular iron restriction was the primary molecular mechanism underpinning the improved erythroid maturation observed in this study, we tested two additional clinically available iron chelators, DFX and DFO, during hematopoietic differentiation of DBA iPSCs. Similar to EPAG, we confirmed that DFX had robust intracellular iron chelating properties in iPSC-derived erythroid cells (Figure 4A,B). In contrast, DFO displayed a more limited direct access to intracellular iron pools (Figure 4C,D), in agreement with the known extracellular iron sink properties of this chelator. Notably, as observed with EPAG, addition of DFX during differentiation promoted erythroid maturation (Figure 4E,F) and colony formation in CFU assays (Figure 4G). In contrast, DFO had a more limited impact on erythropoiesis in this model (Figure 4E–G). Together, these observations suggest that EPAG promotes erythroid maturation by lowering the intracellular iron pools, thereby circumventing the cellular toxicities associated with excessive free heme in DBA erythroid progenitors.

## 4. Discussion

In this study, we combined a reprogramming approach and a scalable monolayerbased hematopoietic differentiation protocol to model ex vivo DBA pathophysiology and test EPAG against disease-specific human cells. We present evidence that addition of EPAG during DBA iPSC differentiation rescues at least in part the block in erythroid maturation recapitulated in this model. These findings and the recent report of a robust erythroid response to EPAG in a steroid-refractory DBA patient with a de novo mutation in RPS19 [42] suggest that EPAG may represent an alternative therapeutic modality for subjects with DBA. A single-arm phase I/II pilot trial (NCT04269889) has recently received approval at our institution to further assess safety and efficacy of EPAG in DBA patients.

Our findings indicate that a molecular mechanism based on intracellular iron chelation accounts for the improved erythropoiesis in our study. Addition of EPAG during iPSC differentiation did not augment non-erythroid lineage output, as measured by CFU assays of hematopoietic cells differentiated from iPSC lines, suggesting a mechanism in- dependent of EPAG’s stimulation of primitive multipotent HSPCs. The robust intracellular iron chelating and mobilizing properties of EPAG have been extensively documented in various cell culture systems [22,23,24,26,41] and in patients treated with this drug [43]. Consistent with this mechanism, we observed that DFX also enhanced erythropoiesis in our DBA iPSC model. Case reports and small studies of patients with transfusion-dependent anemia, including subjects with DBA [44] or other ribosomopathies [45,46], have also suggested that long-term iron chelation therapy with DFX can improve erythroid parameters and reduce transfusion requirements. In contrast, erythropoiesis did not improve considerably in iPSC differentiation cultures supplemented with DFO, a clinically available chelator that only poorly permeates into cells at therapeutic doses [47]. Likewise, notwithstanding that DFO was for decades the only chelating drug available for transfusional hemosiderosis in DBA subjects, no erythroid response has been reported after treatment with this agent [48].

How intracellular iron chelation ultimately improves erythropoiesis in DBA remains to be fully elucidated. As previously proposed [10,12], we posit that iron restriction through chelation with EPAG or DFX may delay heme synthesis and reestablish the aberrant heme/globin ratio to offset cellular toxicities associated with the accumulation of free heme in DBA erythroid progenitors. Consistent with this notion, it was formerly shown that erythroid differentiation of DBA bone marrow cells markedly increased in culture by addition of succinylacetone, a specific inhibitor of heme production [12]. Conversely, in our study, addition of EPAG to control isogenic iPSC cultures partially impeded erythroid differentiation. In the context of normal globin synthesis in erythroid progenitors derived from isogenic iPSC lines, heme is not in excess in the early stages of erythroid maturation and limiting its synthesis by restricting iron pool availability could reduce erythroid output in these cells. Nevertheless, this possibility will require direct validation in the form of quantitative studies comparing heme and globin contents in erythroid cells derived from DBA iPSCs differentiated with or without EPAG. Independent of its effect on heme synthesis, chelation-mediated decrease in redox-active cellular LIP in DBA erythroid progenitors may also promote erythropoiesis by limiting intracellular ROS accumulation and erythroid cell apoptosis [22,26,41,49,50,51]. Similarly, reduction in free intracellular iron has been shown to impair the activity of prolyl hydroxylase domain enzymes, resulting in the stabilization of hypoxia-inducible factor-1 alpha under normoxic conditions, attenuation of injury from ROS and decreased rates of erythroid progenitor cell apoptosis [52,53].

Evidence indicates that the underlying defect in subjects with DBA may not be limited to the erythroid lineage. In the clinic, development of trilineage cytopenias and marrow hypoplasia have been observed, suggesting that the most primitive HSPCs within the bone marrow may also become affected in time by RP haploinsufficiency in DBA [54]. While EPAG stimulation of multipotent HSPCs unlikely contributed to the observed erythroid response in our iPSC model of DBA, trilineage hematopoietic recovery has been documented frequently in SAA patients treated long-term with EPAG in prospective clinical trials [15,16,55] and in a recent practice-based survey [56], suggesting that extended treatment with EPAG might also mitigate the risks of marrow failure in aging DBA patients. Mechanistically, EPAG was shown to bypass the suppressive effect of inflammation [19] and promote DNA repair in HSPCs [21], findings of interest in view of recent experimental data implicating pathways linked to inflammation [57,58,59] and DNA damage [60] in the pathophysiology of DBA. The iron chelating properties of EPAG have also been observed in human HSPCs cultured ex vivo [41], but a causal link between EPAG mediated iron chelation and clinical amelioration of hematologic parameters has not been confirmed [61].

## 5. Conclusions

Our work highlights the ability of iPSC-derived cells to recapitulate in vitro the erythroid phenotype observed in DBA patients, and the potential utility of EPAG to improve erythropoiesis in DBA by restricting iron pool availability (Figure 5). Recent studies evoke the possibility that EPAG might also prevent the development of pancytopenia in aging DBA patients. While concerns have been raised that EPAG could favor selection of pre-malignant HSPCs in cohorts of refractory SAA patients treated with EPAG [15,55], the latent risk of clonal evolution in patients with DBA is unknown. Hence, only carefully monitored prospective clinical trials designed to evaluate safety and efficacy of EPAG in subjects with DBA will ultimately provide irrevocable demonstration of benefit in this disorder.

## Figures and Tables

**Figure 1 cells-10-00734-f001:**
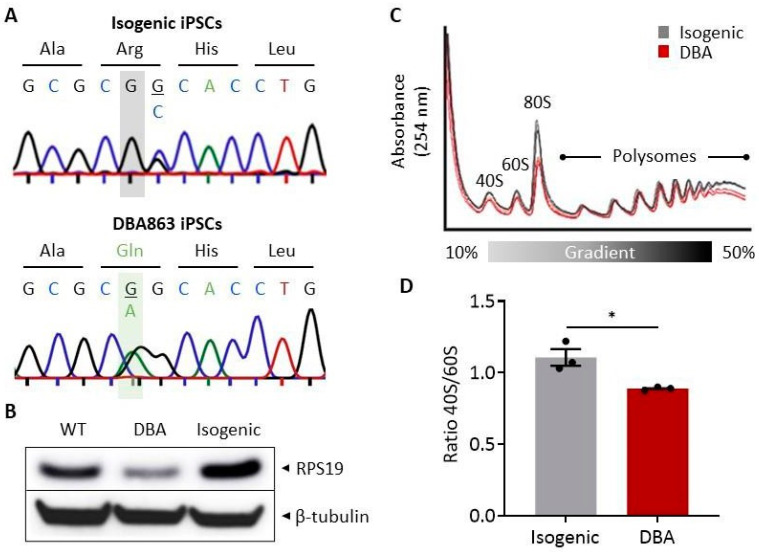
Generation and characterization of isogenic and DBA iPSCs. (**A**) Genomic RPS19 sequence trace chromatograms for isogenic (top panel) and DBA863 (bottom panel) iPSCs to confirm CRISPR/Cas9-mediated correction (grey highlight) of heterozygous c.185G>A (pR62Q) nonsense mutation (green highlight). In isogenic iPSCs, the overlapping G/C bases represent a silent mutation introduced in the template DNA during genome editing to complete an AscI restriction site (GGCGCGCC) for rapid RFLP screening of RPS19 gene corrected iPSC clones. (**B**) Western blot for RPS19 and β-tubulin loading control for wild-type (WT), DBA and isogenic iPSCs. (**C**) Sucrose gradient (10–50%) polysome profiling analyses for isogenic and DBA iPSCs. Peaks represent ribosomal subunits (40S and 60S), 80S monosomes, and ribosome clusters (polysomes). (**D**) Calculated 40S/60S ratios for isogenic and DBA iPSCs. Data are presented as mean ± standard error of the mean (SEM). Unpaired *t*-test, * *p* ≤ 0.05 (*n* = 3).

**Figure 2 cells-10-00734-f002:**
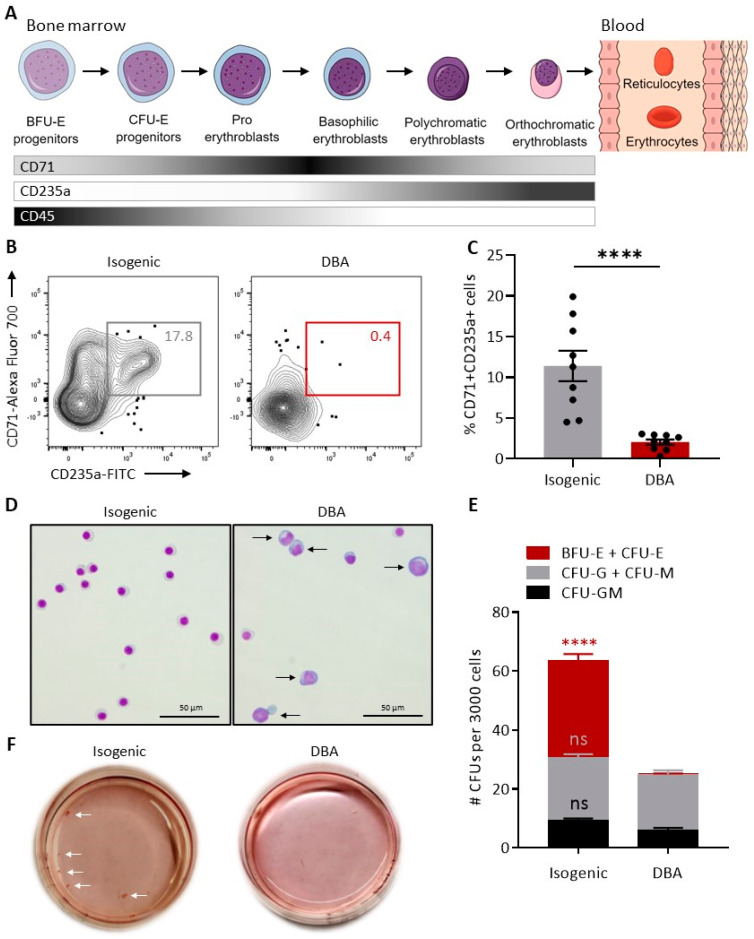
Hematopoietic differentiation of DBA iPSCs phenocopies in vitro the erythroid maturation defect found in DBA. (**A**) Stages of erythroid maturation defined by cell populations expressing variable levels of CD71 (transferrin receptor), CD235a (glycophorin A) and CD45. (**B**) Representative flow cytometry contour plots depicting percentages of late-stage CD45^−^ erythroblasts co-expressing CD71 and CD235a at culture day 21 of isogenic and DBA iPSC differentiation. (**C**) Percentages of CD71^+^CD235a^+^ late-stage CD45^−^ erythroblasts at culture day 19 or 21 of isogenic and DBA iPSC differentiation (*n* = 9). (**D**) Representative Giemsa stains of hematopoietic cells at culture day 21 of isogenic and DBA iPSC differentiation. (**E**) Number of colony forming units (CFUs) per 3000 hematopoietic cells harvested at culture day 19 or 21 of isogenic and DBA iPSC differentiation (*n* = 16). (**F**) Representative CFU plates for hematopoietic cells harvested at culture day 21 of isogenic and DBA iPSC differentiation. White arrows point to erythroid colonies. In panels C and E, data are presented as mean ± SEM. In panel E, statistical analysis is presented for each colony type relative to the “DBA iPSCs” group. Unpaired *t*-test, **** *p* ≤ 0.0001, ns: not significant.

**Figure 3 cells-10-00734-f003:**
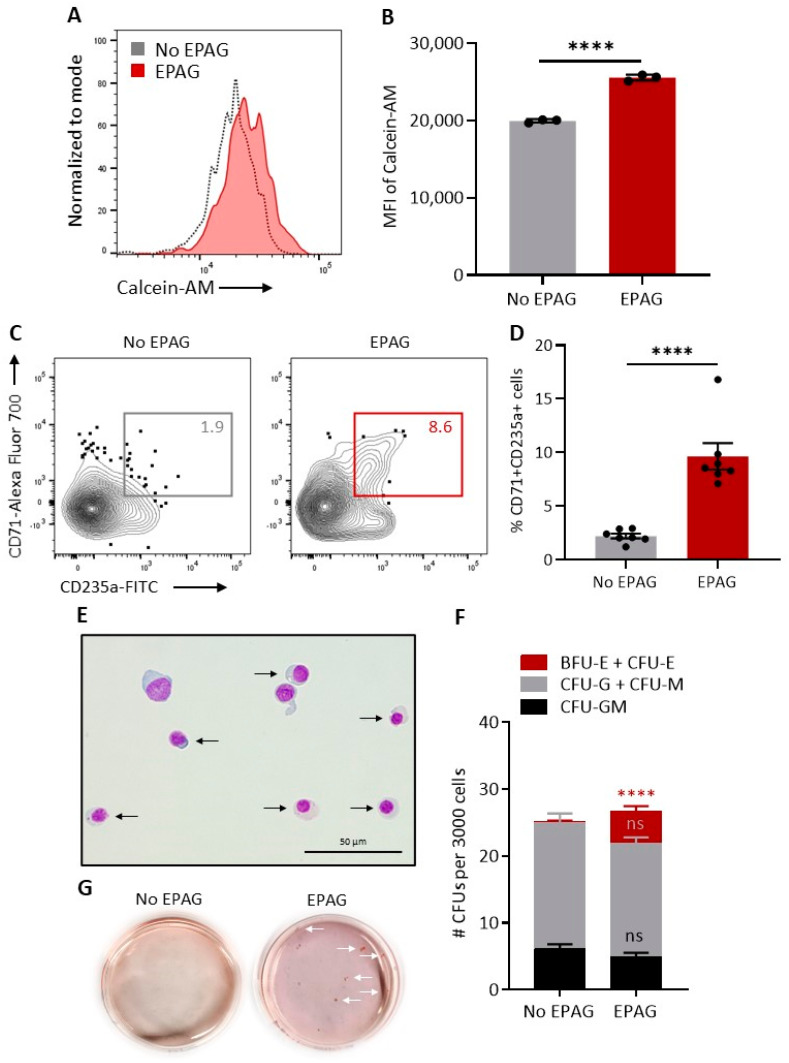
Defective erythropoiesis in DBA iPSCs is partially rescued by eltrombopag (EPAG). (**A**,**B**) Quantification of intracellular iron chelating properties of EPAG in erythroid cells differentiated from isogenic iPSCs. Representative flow cytometry histograms of intracellular calcein-AM fluorescence (A) and summary of the mean fluorescence intensity of calcein-AM (B) in isogenic iPSC-derived erythroid cells loaded with calcein-AM and then treated or not with EPAG. Peak height of histogram was normalized to mode (**C**) Representative flow cytometry contour plots depicting percentages of late-stage CD45^−^ erythroblasts co-expressing CD71 and CD235a at culture day 21 of isogenic iPSC differentiation in the presence or absence of EPAG. (**D**) Percentages of CD71^+^CD235a^+^ late-stage CD45^−^ erythroblasts at culture day 19 or 21 of isogenic iPSC differentiation in the presence or absence of EPAG (*n* = 7). (**E**) Representative Giemsa stains of hematopoietic cells at culture day 21 of isogenic iPSC differentiation in the presence of EPAG. (**F**) Number of colony forming units (CFUs) per 3000 hematopoietic cells harvested at culture day 19 or 21 of DBA iPSC differentiation in the absence or presence of EPAG (*n* = 8). (**G**) Representative CFU plates for hematopoietic cells harvested at culture day 21 of DBA iPSC differentiation in the presence or absence of EPAG. White arrows point to erythroid colonies. In panels B, D and F, data are presented as mean ± SEM. In panel F, statistical analysis is presented for each colony type relative to the “No EPAG” group. Unpaired *t*-test, **** *p* ≤ 0.0001, ns: not significant.

**Figure 4 cells-10-00734-f004:**
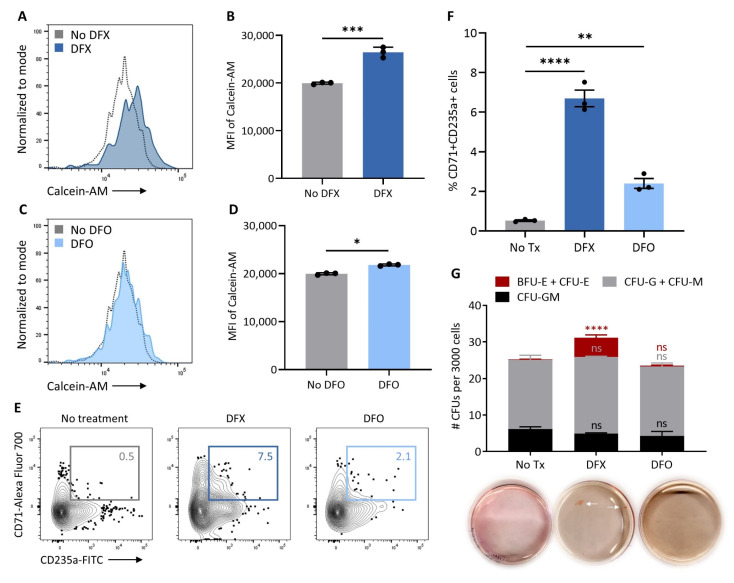
Intracellular iron restriction improves erythropoiesis in DBA iPSCs. (**A**–**D**) Quantification of intracellular iron chelating properties of DFX and DFO in erythroid cells differentiated from isogenic iPSCs. Representative flow cytometry histograms of intracellular calcein-AM fluorescence (**A**,**C**) and summary of the mean fluorescence intensity of calcein-AM (**B**,**D**) in isogenic iPSC-derived erythroid cells loaded with calcein-AM and then treated or not with DFX (**A**,**B**) or DFO (**C**,**D**). Peak height of each histogram was normalized to mode. (**E**) Representative flow cytometry contour plots depicting percentages of late-stage CD45- erythroblasts co-expressing CD71 and CD235a at culture day 21 of DBA iPSC differentiation in in the absence of an iron chelator (no treatment) or in the presence of deferasirox (DFX) or deferoxamine (DFO). (**F**) Percentages of CD71 + CD235a+ late-stage CD45^−^ erythroblasts at culture day 19 or 21 of DBA iPSC differentiation in the absence of an iron chelator (No Tx, no treatment) or in the presence of DFX or DFO (*n* = 3). (**G**) Number of CFUs per 3000 hematopoietic cells harvested at day 19 or 21 of DBA iPSC differentiation in the absence of an iron chelator (No Tx) or in the presence of DFX or DFO (*n* = 4). Insets at the bottom are representative CFU plates; white arrows point to erythroid colonies. In panels B, D, F and G, data are presented as mean ± SEM. In panel G, statistical analysis is presented for each colony type relative to the “No Tx” group. Unpaired *t*-test, * *p* ≤ 0.05, ** *p* ≤ 0.01, *** *p* ≤ 0.001, **** *p* ≤ 0.0001, ns: not significant.

**Figure 5 cells-10-00734-f005:**
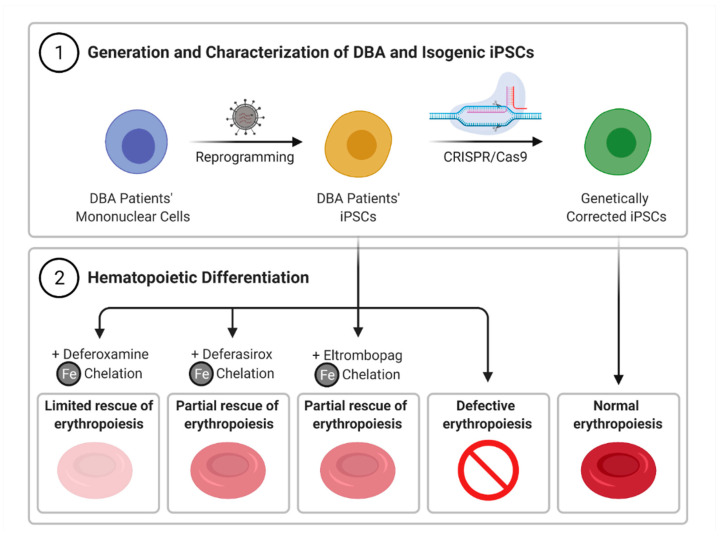
Summary. iPSC lines were generated from blood cells of DBA patients, and isogenic iPSC clones were produced by CRISPR-mediated correction of RPS19 mutations. Hematopoietic differentiation of DBA iPSCs phenocopied the erythroid maturation defect found in DBA. Defective erythropoiesis was partially rescued by eltrombopag and deferasirox, but did not improve substantially with deferoxamine, consistent with a mechanism based on intracellular iron chelation.

**Table 1 cells-10-00734-t001:** RPS19 mutations in subjects with DBA.

Subject ID	Nucleotide Change	Amino Acid Change	Exon
DBA863	c.185G>A	p.R62Q	4
DBA869	c.112A>G	p.K38Q	3
DBA872	c.280C>T	p.R94X	4

## Data Availability

Data that support the findings of this study are available within the article, in the Appendix A and from the corresponding author upon request.

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
