# Peer review of "Eltrombopag Improves Erythroid Differentiation in a Human Induced Pluripotent Stem Cell Model of Diamond Blackfan Anemia"

_cells, 2021, doi:10.3390/cells10040734_

Round 1

Reviewer 1 Report

 This paper need to resubmitted with modification including the results of the following experiments to better characterize the mature erythrocytes.

  1. I would like to see a Flow analysis of cd235vs71 erythrocyic population
  2. Try doing the erythrocyte expansion using D10 of hematopoietic differentiation
  3. Also use D10 cells for CFU assay
  4. Please see the nucleation status of erythrocytes using May Gruntwald-Geimsa staining
  5. Also need to do a Q-PCR of beta globins from erythrocytes.

Reviewer 2 Report

Review:

Here the authors provide a study on iPSC derived from DBA patients to show that eltrombopac improves erythropoiesis by chelating intracellular iron to prevent heme or free iron induced damage due to low globin synthesis during DBA erythropoiesis. This effect could also be observed upon using another intracellular iron chelator, DFX. As there is a dire need for novel treatments to boost erythropoiesis in DBA, besides glucocorticoids, this manuscript provides a novel treatment option. Importantly, eltrombopac is already approved for in human use which could speed-up translation. The report is nicely written, however, there are a few concerns that need to be addressed.

What is missing from the manuscript is the effect of eltrombopac on the syngeneic/healty iPSC erythropoiesis (syngeneic lines) and maybe on CD34+ derived erythropoiesis. This is important to evaluate the effect of intracellular iron chelation (and possibly heme formation inhibitor and globin expression) on erythropoiesis per see. It may not be expected that erythropoiesis upon chelation of intracellular iron will proceed unperturbed. I agree that it may help to tackle the disbalance between heme/iron and globin but titration and dosage will be very important. Is this the reason why the rescue is not complete?

In relation to this, the authors show the effects of a specific concentration of Eltrombopac (and DFX, DFO) but is there a dose response? Does increased concentrations (or reduced for that matter) lead to a better response. Please comment on how and why this particular concentration was chosen for these experiments.

Please comment in the discussion on the other DBA defects in erythropoiesis besides the reduced globin expression and how iron (chelation) may also influence this balance. Taken into account that translation of many RNAs during erythropoiesis are regulated by iron response elements.

Figure 2

Can you please provide a quantification of the total CD71+ population and not only of P1-P3 as this is a measurement of erythroid potential from these iPSC.

Figure 3

What is the effect of EPAG on the repaired iPSC lines? As TPO is an important factor to hematopoietic specification, maintenance and differentiation to lineage, some of the effects may also be due to increased hematopoiesis? This could explain the increased total CD71+ cells from the EPAG treated conditions (as in figure 2 the total CD71+ cells was around 25% also for the repaired line).

Minor:

In figure 2A, basophilic erythroblasts actually still express EpoR normally in adult erythropoiesis. Detecting EpoR by flow cytometry is notoriously difficult, how sure are the authors that the EpoR antibody is actually able to detect lower amounts of EpoR, Was this antibody titrated on adult erythropoiesis from say CD34+ cells? Why did the authors not stick to a more conventional erythropoiesis markers and analysis like CD235 to CD71 or Band3 to CD44 which give excellent resolution within the basophilic-orthochromatic-retic stages that are now defined here as EpoR negative population with reducing CD45 expression but CD71+.

Reviewer 3 Report

The manuscript by Qanash et al. describes the creation and subsequent correction and characterization of Diamond Blackfan Anemia disease-specific iPSCs. Interestingly, and the crux of the work, is the finding that the addition of the TPO mimetic Eltrombopag at least partially rescues the erythoid maturation defect that is a hallmark of the disease. The authors then go on to propose a potential mechanism of action surrounding iron chelation and localization.

General comments:

  1. In general, these are well designed and executed experiments with an accompanying well constructed and easily readable manuscript. Although the creation, correction and characterization of anemia-specific iPSCs is not novel, the application of this model to identify Eltrombopag as a potential therapeutic is interesting and worthy of publication.

  1. This reviewer’s main criticism of the work surrounds the characterization of one of the key findings of the paper: that Eltrombopag improves erythroid maturation in the noted model. More specifically, the authors choose to use the combination of CD71 and EPOR to ‘stage’ the various populations of iPSC-derived erythroblasts that are created. Notably, particularly in a pluripotent stem cell-based system, this is neither comprehensive nor the standard methodology that is usually employed. Unlike in vivo or even primary cell systems, iPSC-based erythroblasts do not significantly downregulate CD71 upon ‘maturation’ and the addition of super-physiological amounts of cytokines like EPO can alter the expression of EPO-R. As such, and perhaps in combination with the markers they have already chosen, the authors should employ the hallmark marker of erythoid specification and maturation GLYA (CD235). This is the standard marker used in the vast majority of studies of this nature. Moreover, to separate different stages of erythropoietic development that may be impacted in their study, the authors should employ the cell surface markers Band 3 and a4-Integrin which allow for the identification and isolation of erythroblasts at specific, successive developmental stages of human erythropoiesis.

  1. A point of consideration that should be addressed in the text: the authors utilize a specification protocol that most likely results in the recapitulation of ‘primitive’ hematopoietic/erythropoietic development. More recent protocols allow for the production of iPSC-derived erythroblasts with an adult-type signature by using early, selective, stage-specific patterning of mesoderm and resultant hemogenic endothelial cells which generate erythroblasts that produce adult hemoglobin. Although it is unclear if this would impact the results of the noted studies, it should be mentioned as a limitation/point of consideration.

Round 2

Reviewer 1 Report

Authors have addressed all the concerns  

Reviewer 2 Report

The authors have addressed the question that were raised to satisfaction.

Reviewer 3 Report

The authors have addressed all of my concerns.